

**Detecting seasonal and long-term vertical displacement in the North**
**China Plain using GRACE and GPS**
Linsong Wang[1,2], Chao Chen[1,2], Jinsong Du[1], and Tongqing Wang[3]
[1]Hubei Subsurface Multi-scale Imaging Lab (SMIL), Institute of Geophysics and Geomatics, China
University of Geosciences, Wuhan, Hubei, China
[2]Three Gorges Research Center for Geo-hazard, Ministry of Education, China University of
Geosciences, Wuhan, Hubei, China
[3]First Crust Deformation Monitoring and Administration Center, China Earthquake Administration,
Tianjin, China
* Corresponding author: Linsong Wang, wanglinsong@cug.edu.cn
**Abstract:**
We employ twenty-nine continuous Global Positioning System (GPS) time series data
together with data from Gravity Recovery and Climate Experiment (GRACE) are analyzed to
determine the seasonal displacements of surface loadings in the NCP. Results show significant
seasonal variations and a strong correlation between GPS and GRACE results in the vertical
displacement component; the average correlation and WRMS reduction between GPS and
GRACE are 75.6% and 28.9%, respectively, when atmospheric and non-tidal ocean effects
were removed, but the peak to peak annual amplitude of GPS (1.2~6.3 mm) is greater than
GRACE-derived (1.0~2.2 mm). We also calculate the trend rate as well as the seasonal signal





caused by the mass load change from GRACE data, the rate of GRACE-derived Terrestrial
Water Storage (TWS) loss (after multiplying by the scaling factor) in the NCP was 3.39 cm/yr
(equivalent to 12.42 km$^3$/yr) from 2003 to 2009. For a 10-year time span (2003 to 2012), the
rate was 2.57 cm/yr (equivalent to 9.41 km$^3$/yr). Basing on spherical harmonic coefficients for
the gravity field and load Love numbers, we use GRACE model to remove the vertical rates
of elastic displacements due to the surface mass change from GPS data. An overall uplift for
the whole region at the 0.04–1.47 mm/yr level from 2004 to 2009, but the rate of change
direction is inconsistent in different GPS stations at -0.94–2.55 mm/yr level from 2010 to

31    2013.

**Keywords:** GPS, GRACE, Seasonal and long-term displacement, Terrestrial water storage,
the North China Plain
**1. Introduction**
Using Global Positioning System (GPS) to monitor crustal motion, especially in the vertical
or height component due to its large amplitude, has been used to study surface loading caused
by mass change. Site-position time series recorded by continuous GPS arrays have revealed
that vertical displacement variations can result from trend or seasonal distribution of mass in a
region or global changes that cause displacement of the Earth's surface (e.g., a change of
continental water [Bevis et al., 2005; van Dam et al., 2007; Wahr et al., 2013], ice [Sauber et
al., 2000; Khan et al., 2010; Nielsen et al., 2013], snow [Heki, 2001; Grapenthin et al., 2006],
ocean [van Dam et al., 2012; Wahr et al., 2014] and atmospheric mass [van Dam et al., 1994;



Boehm et al., 2007]).
On the global scale, terrestrial hydrologic mass exchanges, that causes significant large-scale
loading, occur between the oceans, continents, and atmosphere at seasonal and inter-annual
time scales. On the local scale, inter-annual and long-periodic change in the hydrologic cycle
that most significantly affects loading are large anthropogenic disturbances on groundwater
extraction and artificial reservoir water impoundment and other climate-driven factors (e.g.,
natural floods and droughts) [e.g., Chao et al., 2008; Rodell et al., 2009; Feng et al., 2013;
Joodaki et al., 2014; Wang et al., 2014]. The global-scale mass variations closely related to
changes in terrestrial water storage (TWS) are observed by the Gravity Recovery and Climate
Experiment (GRACE) satellite mission, while the surface elastic displacement can be
estimated if the load and rheological properties of the Earth were known [Farrell, 1972]. The
majority of previous loading studies solved the three components of crustal motion by joint
analysis of GRACE time-variable gravity field coefficients and GPS data [Kusche and
Schrama, 2005; van Dam et al., 2007; Fu and Freymueller, 2012; Fu et al., 2013]. In principle
the loading effects caused by the majority of mass redistributions near the Earth's surface are
from water, atmospheric, and ocean transports on daily to inter-annual timescales [Kusche and
Schrama, 2005]. The variations of the atmospheric and ocean contribution to the surface
displacement can be reasonably modeled, and therefore corrected, using global atmospheric
surface pressure data and space geodetic data, respectively. Thus, after removing the loading
effects of the atmospheric and ocean, GRACE-derived displacement and GPS data allow the
detection of changes in the Earth's larger hydrological storage systems.




In general, changes in TWS capacities depend on precipitation and human consumption.
Variations in TWS may be related to precipitation, which is strongly driven by climate and
can be simulated from global water and energy balance models [Syed et al., 2008]. This is
related to soil texture and root depth in the case of soil water storage (e.g., soil moisture and
vegetation canopy storage can be derived from the Global Land Data Assimilation System
(GLDAS) [Rodell et al., 2004], the WaterGAP Global Hydrology Model (WGHM) [Döll et al.,
2003] and the Community Land Model (CLM) [Oleson et al., 2013]), surface water storage
(e.g., water in rivers, lakes, reservoirs and wetlands can be derived from WGHM while snow
or ice can be derived from WGHM and CLM), and naturally occurring (i.e., climate-driven)
aquifer storage (e.g., groundwater predicted by WGHM and CLM). Variations in TWS may
also be caused by man-made factors, such as water withdrawals for irrigation purposes [Döll,
2009] and dam construction for power generation and navigation [Wang et al., 2011]. These
changes in TWS can be observed in situ (i.e., groundwater level and impounded water level).
All can cause variations in TWS can lead to the overall changes in crust displacement.

This study focuses on the crustal deformation of the North China Plain (NCP) (Figure 1),
which is one of the most uniformly and extensively altered areas by human activities in the
world [Tang et al., 2013]. The NCP is one of the world's largest aquifer systems and supports
an enormous exploitation of groundwater. Overexploitation of groundwater has seriously
affected agriculture irrigation, industry, public supply, and ecosystems in the NCP. Previous
studies used GRACE data, land surface models, and well observations to provide insight on



groundwater depletion in the NCP [Feng et al., 2013; Moiwo et al., 2013; Tang et al., 2013;
Huang et al., 2015]. Liu et al. [2014] has discussed loading displacement in the NCP before.
Only five GPS stations (i.e., BJFS, BJSH, JIXN, TAIN, and ZHNZ) data are used in their
work. Although they calculated the seasonal amplitudes, phases and trends of vertical
displacement from GRACE and GPS, the atmospheric and non-tidal ocean loading effects
were not removed in the Liu et al.'s work, i.e., added the Atmosphere and Ocean De-aliasing
Level-1B (AOD1B) solution (GAC solution) back to the GRACE spherical harmonic
solutions.

Here, we use GRACE and data from 29 GPS sites to study the seasonal and long-term loading
displacement due to dynamic hydrological processes and groundwater-derived land
subsidence in the NCP. In contrast to previous focus study [Liu et al., 2014]. the most obvious
difference between our results and their work is we removed other loading effects (e.g.,
atmospheric and non-tidal ocean) in order to reflect the seasonal and long-term displacement
caused by TWS loads better. Additionally, we discuss long-term trend due to mass change
revealed by GRACE measurements and its impacts on tectonic vertical rates evaluations.

**2. Data Analysis**
**2.1 GRACE Data**
The GRACE mission design makes it particularly useful for time-variable gravity studies.
GRACE was jointly launched by NASA and the German Aerospace Center (DLR) in March
2002 [Tapley et al., 2004a]. The Level-2 gravity products consist of complete sets of spherical



harmonic (Stokes) coefficients out to some maximum degree and order (typically $l_{max} = 120$)
averaged over monthly intervals. When considering large-scale mass redistribution in the
Earth system on a timescale ranging from weekly to interdecadal, it is reasonable to assume
that all relevant processes occur in a thin layer at the Earth's surface [Kusche and Schrama,
2005]. In this analysis, we assume that the gravitational and geometrical response of the Earth
can be described by Farrell's [1972] theory, where the loads Love numbers only depend on
the spherical harmonic degree. Thus, the elastic displacements due to the surface mass change
can easily be represented in terms of spherical harmonic coefficients for the gravity field and
load Love numbers, $k_l$, $l_l$, and $h_l$ [Wahr et al., 1998; Kusche and Schrama, 2005]. Level-2
products are generated at several project-related processing centers, such as the Center for
Space Research (CSR) at the University of Texas, GeoForschungsZentrum (CSR) in Potsdam,
Germany, and the Jet Propulsion Laboratory (JPL) in California. The mass estimates (TWS
and sea level) show very good agreement among these products [Fu and Freymueller, 2012;
Wahr et al., 2014; Wang et al., 2014].

This study used monthly sets of spherical harmonic (Stokes) coefficients from GRACE RL05
(i.e., Release 5) gravity field solutions generated from the CSR, spanning from February 2003
to April 2013. Each monthly GRACE field consisted of a set of Stokes coefficients, $C_{lm}$ and
$S_{lm}$, up to a degree and order ($l$ and $m$) of 60. In fact, the GRACE Stokes coefficients ("GSM"
coefficients denoted by the GRACE Project) have had modeled estimates of the atmospheric
and oceanic mass signals removed. Thus the GRACE coefficients include the full terrestrial
water storage signal with remaining atmospheric and oceanic signals due to errors in the





respective models [Swenson et al., 2008]. Generally, using the GRACE AOD1B products can
add back the de-aliasing atmospheric and non-tidal oceanic effects to the GRACE data.
However, we would like to reduce the environmental loading contributions to the GRACE
and GPS observations, if we study on the accurate interpretation of displacement due to TWS
loading. Thus, we analyzed on the effects of non-tidal ocean variations and atmospheric
loading on the GRACE model and GPS coordinates, please see Section S1 in the supporting
information for details.

We replaced the GRACE $C_{20}$ coefficients with $C_{20}$ coefficients inferred from satellite laser
ranging [Cheng et al., 2013]. Due to the fact that the reference frame origin used in the
GRACE gravity field determination is the Earth's center of mass (CM), GRACE cannot
determine the degree-1 terms variations in the Earth's gravity field (geocenter motion). Here,
we used degree-1 coefficients as calculated by Swenson et al. [2008] to determine the position
of the CM relative to the center of figure (CF) of the Earth's outer surface. We applied the
post-processing method described in Swenson and Wahr [2006] to remove north-south stripes.
We adopted 250 km as the averaging radius to implement Gaussian smoothing, a technique
which suppresses errors at high degrees [Wahr et al., 1998; van Dam et al., 2007]. Stokes
coefficients results from A et al. [2013] were used to remove contributions from Glacial
Isostatic Adjustment (GIA). The contribution of GIA is about 0.28–0.33 mm/yr and
non-seasonal in the NCP, which is small and non-seasonal; so their impact on the seasonal
results discussed in this paper would be minimal, even if they were not removed.
The spatial pattern of TWS, shown in Figure 2, was obtained from monthly GRACE mass





solutions for NCP and surrounding regions between spring, 2003, and spring, 2013. An
obvious negative trend was identified localized over North China, including some of the
Northwest regions (i.e., Shanxi province) and Northeast regions (i.e., Liaoning province). The
TWS changes derived from the GRACE data show significant loss trends across the entire
study area (NCP), specifically in Beijing, Tianjin, Hebei province, and Shanxi province.
Previous studies have investigated how much groundwater depletion has caused the
GRACE-derived TWS loss in the whole of the NCP [Feng et al., 2013; Moiwo et al., 2013] or
in different sub-regions of the NCP [Tang et al., 2013; Huang et al., 2015]. These
investigations, however, did not focus on regional displacement due to seasonal or long term
variations of hydrologic loading.

**2.2 GPS Data**
Twenty-four GPS sites from the Crustal Movement Observation Network of China
(CMONOC) and five GPS sites from the International GNSS Service (IGS) (Table 1) were
analyzed in this study (Figure 1 shows the locations of the GPS stations). Eight GPS sites of
them were located in the surrounding area of the NCP. Daily values for the upward, eastward
and northward coordinates were determined by GPS data of IGS stations between 2003 and
2013, which is consistent with GRACE time span. The 24 GPS sites of CMONOC provided
data from 2010 to 2013. GIPSY/OASIS-II (Version 5.0) software was used in point
positioning mode to obtain the daily coordinates and covariances; these were used to
transform the daily values into ITRF2008 [Altamimi et al., 2011]. We estimated this daily
frame alignment transformation using a set of reliable ITRF stations (~10 stations each day).





In the GPS processing, corrections for solid Earth tides were undertaken, and ocean tide
loading effects were corrected using ocean tide model FES2004 with Greens Functions
modeled in the reference frame of CM (center of the mass of the total Earth system) to
maintain theoretical consistency and adherence to current IERS conventions [Hao et al., 2016;
Fu et al., 2012], but atmospheric pressure loading or any other loading variations (non-tidal
ocean loading) with periods > 1 day were not removed. In order to focus on the seasonal and
trend feature over the entire observation time, we first smoothed the data to reduce large
scatter before using a 3-month-wide moving window filter to remove the short-period terms.

Due to the coseismic displacement of the 2011 Mw9.0 Tohoku earthquake, we estimated and
removed offsets (i.e., using the differences of the average values of seven days between
before and after earthquake to obtain the coseismic displacement) at those times for the
vertical time series of GPS stations at Eastern China. Wang et al. [2011] study results reveal
that the coseismic horizontal displacements induced by the earthquake are at the level of
millimeters to centimeters in North and Northeast China, with a maximum of 35 mm, but the
vertical coseismic and postseismic displacements are too small to be detected. In order to
maintain consistency with the GIA effects present in the GRACE solutions, we remove GIA
effects for all GPS stations using Stokes coefficients ($l$=100) results computed by A et al.
[2013], which used the ICE5G ice history and VM2 viscosity profile [Peltier et al., 2004].

Figure 3a shows the time series (2003-2012) of daily solutions for IGS GPS sites BJFS,
ZHNZ, BJSH, JIXN and TAIN. The long-term linear trends are mainly dominated by surface



mass loading and tectonic processes, and the GPS time series shows significant seasonal
variations. The peak-to-peak seasonal amplitude can be seen to be more than 20 mm which
reflects the strong seasonal mass changes in the NCP. The GRACE data from CSR uses model
output to remove the gravitational effects of atmospheric and oceanic mass variability from
the satellite data before constructing monthly gravity field solutions. In order to compare the
displacement from GPS with GRACE, the effects of atmospheric and non-tidal oceanic
loading on the GPS coordinates needed to be removed. Displacements due to atmospheric
loading were calculated using data and programs developed by the GGFC (Global
Geophysical Fluid Center) (T. van Dam, NCEP Derived 6 hourly, global surface
displacements at 2.5° ×2.5°  spacing, http://geophy.uni.lu/ncep-loading.html, 2010). These
utilized the NCEP (National Center of Environmental Protection) reanalysis surface pressure
data set. The 12-hour sampling model, ECCO (Estimating the Circulation & Climate of the
Ocean, http://www.ecco-group.org/), is used to compute the surface displacement driven by
non-tidal ocean effects and its spatial resolution is 1 °×0.3-1.0 °, i.e., 1 degree longitude (zonal)
interval and 0.3 to 1.0 degree in latitude (meridian) intervals from equator to high latitude. An
example of the effects of the non-tidal ocean and atmospheric loading in the GPS and
GRACE data is provided in the supporting information (Figure S1).

The displacements caused by atmospheric pressure and non-tidal ocean loading mainly show
seasonal fluctuations and no obvious long-term trend during GPS observation (e.g., time
series of height from atmospheric and non-tidal ocean loading at IGS sites in Figure 3b). The
annual amplitude is 4.0–4.6 mm and 0.24–0.42 mm for the atmospheric and non-tidal ocean





loading effects, respectively, while the semi-annual amplitude is about 0.3 mm and 0.03 mm,
respectively. But the phases between the atmospheric and non-tidal ocean loading effects have
more apparent difference. The results of the seasonal amplitudes and phase fits of vertical
displacements, derived by GRACE and GPS for IGS stations between before and after
corrected atmospheric and non-tidal ocean, are summarized in Table S1 in the supporting
information.

**2.3 Elastic Displacements Due to Mass Loads**
GRACE Stokes coefficients [Wahr et al., 1998] and load Love numbers [Farrell, 1972] can be
used to estimate the displacement effects in three components (Up, North and East) caused by
mass load changes. The mathematical relationships [Kusche and Schrama, 2005; van Dam et
al., 2007] between the radial surface displacement (Up or Height) and the Stokes coefficients
of mass is:
$$\Delta h = dr(\theta, \phi) = R \sum_{l=1}^{N_{max}} \sum_{m=0}^{l} \tilde{P}_{l,m}(\cos\theta) \cdot (C_{lm}\cos(m\phi) + S_{lm}\sin(m\phi)) \frac{h_l}{1+k_l} \qquad (1)$$
where $\Delta h$ is the displacement of the Earth's surface in the radial direction at latitude $\theta$ (theta)
and eastward longitude $\Phi$ (phi); $N_{max}$=60, $R$ is the Earth's radius; $\tilde{P}_{l,m}$ is fully normalized
Legendre functions for degree $l$ and order $m$; $C_{lm}$ and $S_{lm}$ are time variable components of the
$(l,m)$ Stokes coefficients for some month; and $h_l$, $k_l$ and $l_l$ are the three degree dependent load
Love numbers which are functions of Earth's elastic property. In this equation we adopted the
load Love numbers provided by Han and Wahr [1995].

Similarly, horizontal displacements (North and East) can be calculated using the following





equations:
$$\Delta n = dr(\theta,\phi) = -R \sum_{l=1}^{N_{max}} \sum_{m=0}^{l} \frac{\partial}{\partial\theta}\widetilde{P}_{l,m}(\cos\theta)\cdot(C_{lm}\cos(m\phi)+S_{lm}\sin(m\phi))\frac{l_l}{1+k_l}, \qquad (2)$$

$$\Delta e = dr(\theta,\phi) = \frac{R}{\sin\theta} \sum_{l=1}^{N_{max}} \sum_{m=0}^{l} \widetilde{P}_{l,m}(\cos\theta)\cdot m(-C_{lm}\sin(m\phi)+S_{lm}\cos(m\phi))\frac{l_l}{1+k_l}, \qquad (3)$$

where $\Delta n$ and $\Delta e$ are north and east components of the displacement, respectively, with both
having positive values when the crust moves towards the north and east, respectively. As is
mentioned in Section 2.1 above, in order to be consistently comparable to the GPS time series,
we corrected the degree-1 components to GRACE-derived mass variations, using Stokes
coefficients derived by Swenson et al. [2008]. With corresponding to degree-1 contribution to
vertical displacement, the value of load Love numbers of the degree-1 in the CF frame should
be computed by using equation (23) in Blewitt [2003].

Figure 4 shows an example (site BJFS, JIXN, TAIN and ZHNZ) of the GRACE-derived
vertical (Figure 4c) and horizontal displacements (Figure 4a and 4b) before and after
destriping. It can clearly be seen that the maximal amplitude of vertical displacement is
several order of magnitude higher than horizontal displacements. In addition, the calculated
results using the monthly GRACE model data after destriping show that the effects of TWS
(soil moisture, etc.) on surface displacements are seasonal variations and long-term changes
on vertical and horizontal components. As most of the stations are located in areas of TWS
loss in the NCP (see sites location in Figure 2), the fact is that the motion is upward (see the
positive trend of GRACE-derived vertical in Table 1) during this event (if a load is removed,
the site uplifts and moves away from the load [Wahr et al., 2013]). Identified horizontal





displacements are important as they constrain the location of load changes [Wahr et al., 2013;
Wang et al., 2014]. The displacement of the ZHNZ site is upwards and to the south (see the
negative trend of the ZHNZ north component in Figure 4a) due to the mass loss almost due
north of the site. Correspondingly, the displacement of the TAIN site is upwards and to the
southeast (see the negative trend of the north component and the positive trend of the east
component of the TAIN site in Figure 4a and 4b) caused by the mass loss located to the
northwest of the site, based on the use of GPS horizontals for loading studies from Wahr et al.
[2013].

**3. GRACE-derived Seasonal Variations and Comparison with GPS Measurements**
Using equation (1) and GRACE-derived Stokes coefficients, the vertical displacements at the
GPS sites in the NCP and its surrounding region can be calculated. To focus on these changes,
GRACE-derived vertical displacements were computed by fitting a model with a linear trend
and annual periodic terms using Least-Squares method over the entire 11 year time span, for
comparison to the seasonal variations observed by GPS (Table 1). Figure 5 shows time series
of vertical displacements for GPS sites of IGS stations (BJFS, BJSH, JIXN, TAIN and ZHNZ).
The fitting results show the GRACE-derived (without ADO1B) peak-to-peak annual
amplitudes can be more than 2 mm, and the semi-annual amplitude are also visible at these
five GPS sites. This reflects the climate-derived seasonal hydrological fluctuations in the
NCP.

Compared with GRACE results mainly due to the mass change in seasonal and long-term





linear period, all GPS time series show significant seasonal and long-term trends which are
mainly dominated by tectonic and hydrological process. The fitting results (after
Least-Squares fitting) show the peak-to-peak vertical seasonal displacements from GPS time
series to be larger than GRACE-derived results at those GPS sites, and the peak-to-peak
seasonal amplitude changes between 5 mm and 6 mm (Table 1). The results of the comparison
between GPS and GRACE-derived seasonal height variations at 24 GPS sites from
CMONOC can be seen in Figure S3 in the supporting information. For all the selected GPS
sites, the annual component is more dominant than the semi-annual one. The peak-to-peak
annual amplitude is 1.2~6.3 mm and 1.0~2.2 mm for the GPS and GRACE solutions,
respectively, while the semi-annual amplitude is about 1/2~1/3 times of that in annual
amplitude. These more consistent seasonal variations of GRACE and GPS height time series
reflect the climate-derived seasonal hydrologic process, i.e., heavy monsoonal precipitation in
the late summer months result in mass loads increase (the maximum negative of vertical
amplitude) and largely pumped for agricultural usage in late spring months cause mass loads
decrease (the maximum positive of vertical amplitude). We observe the fact that the amplitude
of GPS is relatively larger than that of GRACE, it is not only exists in the IGS stations, but in
almost all stations except SXGX (Table 1). This indicates that GPS has a strong sensitivity for
local surface loading. By contrast, because the spatial resolution of GRACE data is limited to
approximately 300 km ($N_{max}$=60), GRACE-derived results are mainly constrained by large
scale areas. This means that GRACE-derived vertical displacements show a small difference
between stations due to the results are averaged over scales of several hundred km or more.





As is mentioned above, the cause of the difference between our results and Liu's work [Liu et
al., 2014] is we removed atmospheric and non-tidal ocean loading effects while they did not.
However, we found that the amplitude of GPS after removed atmospheric and non-tidal ocean
loading effects, is still greater than the GRACE while we added the AOD1B de-aliasing
model to the GRACE solutions (i.e., no atmospheric and non-tidal ocean corrected, please see
the Table S1 in the supporting information). The most obvious difference between our results
and Liu's work [Liu et al., 2014] is that they adopt the load Love numbers from Guo et al.
[2004] to transform these coefficients into vertical surface displacement estimates. We check
the two results of Love numbers (ocean-load and atmospheric pressure-load) from Guo et al.
[2004], there are significant differences between ocean-load and atmospheric pressure-load
Love numbers. Meanwhile, we compared $k_n$ Love numbers from Guo et al. [2004] (Liu et al.'s
work) and $k_n$ Love numbers from Han and Wahr [1995] (our work) with the $k_n$ Love numbers
used in ADO1B products [Farrell, 1972], respectively. The different Love numbers have
caused the amplitude of the same station from Liu's GRACE-derived vertical displacements
much more than GPS and our GRACE results, due to $k_n$ from atmospheric pressure-load Love
numbers [Guo et al., 2004] significantly larger than Love numbers from Han and Wahr [1995]
and Farrell [1972]. The detailed analysis of the different Love numbers from Guo et al. [2004],
Han and Wahr [1995] and Farrell [1972], please see the Section S1 in the supporting
information

Next, we compare GPS observed and GRACE-derived seasonal height variations. The
estimated annual amplitudes and initial phases derived from GPS (grey vector) and GRACE




(red vector) are shown in Figure 6. We find that there are many sites where the signals
disagree in both amplitude and phase. The annual amplitudes and phases from
GRACE-derived results are much more spatially coherent than those determined from the
GPS heights. Because GRACE solutions truncate to $l_{max}$=60 and the Gaussian filtering was
used to lead to so smooth out concentrated loads. The phases of the GRACE data show that
the annual signal peaks (summer monsoon) basic between September and October and
indicate that the maximum load occurs in this two months. But there are several sites where
the GRACE signals disagree in phase with GPS data, which the annual signal peaks sometime
between August and September. The five signals of sites in the northwest foothills region of
NCP agree in phase, while annual amplitudes from GRACE are significantly less than GPS,
e.g., NMTK, NMZL, HEYY, HEZJ and HECC. The cause of mostly phase inconsistency may
be the different spatial resolution of GRACE compared to GPS. That is, GPS measurements
can sense the difference between loads very near the site, and loads a bit further away, but
GRACE with wavelengths on the order of 300 km reflects this variation at a monthly scale.
Another important reason is that a one-month sampling of GRACE means a phase sampling
of 30°, while a one-day sampling of GPS means a phase sampling of ~1°, the different
temporal sampling rate caused the inconsistent phase between GRACE and GPS.

With the purpose of quantitatively evaluating the consistency between GPS and GRACE, we
remove GRACE-derived seasonal displacement from GPS observed detrended height time
series to compute the reductions of Weighted Root Mean Squares (WRMS) based on the
equation (2) in van Dam et al. [2007]. Correlation between GPS and GRACE derived





seasonal variations and WRMS reduction ratio of remove GRACE-derived seasonal
displacement from GPS observed detrended height time series, please see the Table S4 in the
supporting information for details. All the selected sites show high correlation (85%−99%,
without TJBH site) when atmospheric and non-tidal ocean effects was not removed. Our
correlation results of IGS stations (BJFS, BJSH, JIXN, TAIN and ZHNZ) are consistent with
Liu's work [Liu et al., 2014], indicating that the seasonal variations might come from the
same geophysical process. The WRMS residual reduction ratio for all the stations ranged
from 19% to 85%, which is better than Liu's work [Liu et al., 2014]. However, the correlation
and WRMS reduction between GPS and GRACE are weak when atmospheric and non-tidal
ocean effects was removed, with the average correlation and WRMS reduction reduce to 75.6%
and 28.9%, respectively. This is mainly because the seasonal hydrologic process is major
contributors to seasonal changes, and different stations are greatly influenced by the
surrounding hydrological process. By contrast, the seasonal amplitudes and phases from
GRACE results are much more spatially coherent than those determined from the GPS
heights, caused by the different spatial resolution between them. In addition, we also attempt
to calculate GRACE-derived horizontal displacement using equation (2) and (3), and compare
it with the GPS measurements. An example (five IGS sites) of the comparison between
GRACE-derived and GPS observed horizontal displacements were presented to demonstrate
the correlation of seasonal horizontal variation caused by surface hydrological load. Please
see the Figure S4 in the supporting information.

**4. Long-Term Uplift Caused by TWS Loss**



To estimate TWS changes averaged over the NCP, an averaging kernel based on weighted
Gaussian convolution to construct monthly time series from GRACE Stokes coefficients
described by equation (5) of Wahr et al. [2014] was used. This method extends the averaging
kernel convolution approach [Swenson and Wahr, 2002] by allowing for nonuniform
weighting during the convolution. We took the NCP "basin function" from the China
provincial boundary grid points and we convolved with a 250 km Gaussian smoothing. We
then applied this averaging kernel to GRACE Stokes coefficients to obtain a TWS time series
for NCP (Figure 7). The results identified a continuous decrease in TWS from 2004 to 2009;
the rate of this decrease slowed towards the end of 2009. The rate of TWS loss obtained by
this analysis was 1.62 cm/yr from 2003 to 2009 and 1.23 cm/yr from 2003 to 2012 (Table 2).

The estimated results for the time series analysis also include some contributions outside the
NCP due to the finite number of harmonic degrees in the GRACE solution (e.g., $l_{max}$=60 for
CSR solutions). The average kernel in our study is also not an exact unity cover for the entire
NCP area; these two factors result in under or overestimation of the true TWS time series
signal. To estimate this "leakage in" signal, a scaling factor method was used to restore the
amplitude-damped TWS time series. This method, as described by Wahr et al. [2014], requires
the construction of a set of simulated Stokes coefficients which represents the signal from a
uniformly distributed 1 cm water depth change over the NCP. This estimates a water
volume=3.6626 km$^3$ based on the overall area of "basin function" (i.e., 366260 km$^2$). By
applying our GRACE analysis procedure to these simulated Stokes coefficients, we can infer
an average water thickness change equal to 0.47 cm for the NCP. Each monthly GRACE





estimate of NCP water thickness is then multiplied by a scaling factor=1 (cm)/0.47 (cm) to
obtain variations in the total water thickness per area of the NCP. Multiplying the monthly
GRACE estimates of NCP water thickness by a scaling factor=3.6626 (km$^3$)/0.47 (cm)
provides a mass change of the NCP. Table 2 shows the rate of GRACE-derived TWS loss
(after multiplying by the scaling factor) in the NCP was 3.39 cm/yr from 2003 to 2009; this is
equivalent to a volume of 12.42 km$^3$/yr. For a 10-year time span, the rate was 2.57 cm/yr,
which is equivalent to a volume of 9.41 km$^3$/yr.

Loading or unloading of the crust from surface mass changes will cause the crust to subside or
uplift with different amplitudes. These displacements depend on the amplitude of the load and
the distance between the load and the observation point [Farrell, 1972]. On this basis, we used
GRACE-derived vertical displacements (the method of elastic displacements due to mass
loads described by Section 2.3) to evaluate TWS loss contributions for the evident crustal
uplift in the GPS measurements. Time series of monthly predicted vertical surface
displacements from GRACE for 25 GPS sites in the NCP were plotted (Figure 8a). The fitting
results (after Least-Squares fitting) show the trend rate of GRACE-derived vertical
displacements for the whole region at the 0.37–0.95 mm/yr level from 2004 to 2009, but the
rate of change direction is inconsistent in different GPS stations at -0.40–0.51 mm/yr level
from 2010 to 2013 (Table 1). The smoothed results indicate a rising trend from 2004 to 2012
(Figure 8a) which represented the TWS loss in the observation time span. Figure 8a also
clearly shows mass anomaly due to TWS changes in the vertical component, e.g., a notable
negative peak from 2003 to 2005 and subsidence in 2012 (grey background in Figure 8a).



These GRACE-derived long-term height fluctuations mainly include variations in the storage
of natural surface water: high storage in wet years and low storage in dry years [Tang et al.,
2013], which can be modeled using land surface model output such as those provided by the
GLDAS [Rodell et al., 2004]. We clearly see that these fluctuations are almost erased by
removing the modeled soil moisture (SM) of GLDAS/Noah, and the obvious uplift are
presented in the decomposition of the signal (Figure 8b), which is mainly because the
contributions from groundwater depletion in the NCP (please see the section 5.1 for discuss in
details).

**5. Discussion**
**5.1 Groundwater Depletion Contributions to Long-Term Uplift**
The GRACE-derived vertical displacements are also the effect of mass loading sensitive to
water at all depths: surface water storage, soil moisture, snow and groundwater, including
anthropogenic effects (i.e., groundwater withdrawal, inter-basin diversion, reservoir and coal
transport). To isolate the groundwater contributions, the Noah version of GLDAS which
possesses monthly intervals and spatial resolution of 1.0 degrees [Rodell et al., 2004] was
used to subtract monthly water storage estimates predicted by land surface models. GLDAS
generates a series of land surface forcing (e.g., precipitation, surface meteorology and
radiation), state (e.g., soil moisture and temperature, and snow), and flux (e.g., evaporation
and sensible heat flux) data simulated by land surface models. The GLDAS/Noah model can
provide values of snow, vegetation and all soil moisture layers, but it does not include
anthropogenic and climate-driven groundwater depletion. So groundwater contributions





retained in heights time series when GRACE-derived vertical displacements subtracting
GLDAS/Noah effects.

Figure 8b shows the GRACE-derived height amplitudes after using output from the
GLDAS/Noah hydrology model to remove the continental water storage signal. The
calculated results show that the contributions of other types of TWS effects (except
groundwater) on the surface are small relative to groundwater depletion, and those loads
mainly effects on the amplitudes of seasonal displacement with no obvious long-term trend.
Compare the seasonal amplitudes, phases and trend fit of vertical displacements between
before and after remove GLDAS/Noah effects from GRACE-derived displacement, please see
the Table S4 in the supporting information.

For the results described above, after the subtraction of the GLDAS/Noah contributions,
GRACE-derived heights largely reflect loading effects from the groundwater (natural and
anthropogenic factors) and anthropogenic contributions. The anthropogenic impact on mass
change was investigated by Tang et al. [2013] for the effect of inter-basin diversion, reservoir
and coal transport distribution on the GRACE-derived estimates of groundwater depletion in
the NCP. Results from their investigation showed that the trend of anthropogenic
contributions was equivalent to 4.83 mm/yr water thickness (described by equations in Table
2 of Tang et al. [2013]) during 2003-2011 for the whole NCP. This means that there was a
large groundwater depletion contribution for the GRACE-derived vertical displacements in
long-term uplift. Investigating groundwater withdrawal due to anthropogenic activities





(drinking water extraction, agricultural irrigation and industrial manufacturing) should be of
high importance because precipitation data for this area (provided by the China
Meteorological Data Sharing Service System, available at http://cdc.nmic.cn/home.do)
indicated no long-term droughts during the GRACE observation period of 2003-2011.

**5.2 Removing Hydrological Loading Displacement from GPS Using GRACE Data**
Coordinate variations measured by GPS stations, principally for the vertical component, have
been used to investigate global [Dong et al., 2002] and local [Grapenthin et al., 2006] tectonic
activity, as well as seasonal displacement modes for constraining estimates of continental,
atmospheric and ocean water storage. Some previous studies [e.g., Fu et al., 2012] have
focused on the vertical component of crustal motion with relying on the accurate
interpretation of GPS motion in terms of surface stress or tectonic movement. Thus, the
displacement signal from surface mass loading is a source of noise [van Dam et al., 2007].
For these applications, they would like to obtain reliable loading models or even surface mass
observations, which can be used to reduce the environmental loading contributions to the GPS
observations. In this study, we also attempt to separate tectonic and hydrological effect using
GRACE-derived hydrological vertical rates. As mentioned in the Section 3, the good seasonal
correlation between GRACE and GPS signals indicates that the long-term uplifts revealed by
GRACE detections are probably true and mixed in the GPS measurements.

Figure 9 shows results for individual GPS time series. Crustal subsidence or uplift due to
vertical tectonic motion and TWS changes in the studied period are clearly evident in the





vertical component shown in most of the GPS stations; In fact, the analysis in the five IGS
GPS stations (BJFS, BJSH, JIXN, TAIN and ZHNZ) suggests that the GPS vertical time
series can be described by two different rates around 2010, due to a continuous decrease in
TWS from 2004 until 2009; towards the end of 2009 the rate of this decrease slowed and rate
started to rise since 2010 (please see Figure 6). Thus, we divided it into two periods when
fitting GPS and GRACE trend for these five stations (Figure 9a). GPS trend changes indicate
an overall uplift for the whole region at the 0.04–1.47 mm/yr level from 2004 to 2009, but the
rate of change direction is inconsistent in different GPS stations at -0.94–2.55 mm/yr level
from 2010 to 2013 (Table 1).

In addition, the long-term trend rate is different in different areas from 2010 to 2013 (Figure
9b). For example, the trend rate from GPS measurements shows the uplift in western NCP
(Shanxi-SX,and some of Hebei-HE stations), but opposite trends in the central and eastern
plain of NCP (Beijing-BJ, Tianjin-TJ and some of Hebei-HE stations). The groundwater
depletion which occurs in the shallow unconfined aquifers in Piedmont Plain leads to the
loading uplift effect from mass loss. But groundwater depletion occurs in the deep confined
aquifers in the central and eastern plain of NCP, which causes serious ground subsidence,
rather than ground uplift caused by groundwater loss.

It is also possible that some GPS signals could be a result of loading from changes in the
distribution of water stored in the surface and ground around the GPS surrounding region. To
remove those contributions, Stokes coefficients output from the GRACE model were used to



compute crustal motion at the NCP (Figure 8a), and then transform the monthly results into
daily resolution data using a spline interpolation. In addition, GRACE solutions are corrected
for GIA while GPS ones are not. Here Stokes coefficients results from A et al. [2013] were
used to remove GIA effects from GPS measurements, which is about 0.2–0.4 mm/year in the
land areas of China and 0.28–0.33 mm/year in the NCP (Please see the Figure S5 and Table
S4 in the supporting information).

We compute the GRACE-derived long-term uplift for all continuous GPS sites used in this
paper. The results indicate an overall uplift for the NCP region. Then we remove this TWS
induced uplift from GPS actual observed vertical rates to derive the corrected vertical
velocities. Figure 10 divided the time into two periods (2004–2009 and 2010–2013) to
indicate an overall long-term trend before (gray arrow) and after (red arrow) removing
hydrological loading displacement for the whole region. Secular displacement results between
2004 and 2009 show that loading displacement due to the TWS loss reduce the uplift rate of
GPS to some extent, and groundwater exploitation was the main contributor to crustal uplift
caused by TWS loss in the NCP (BJFS, BJSH and JIXN in the Figure 10a). However, studies
indicate that groundwater withdrawal produces localized subsidence which can be largely
relative to tectonic displacement [Bawden et al., 2001]. Therefore, in this study, more
attention was paid to land subsidence due to groundwater loss.

**5.3 Land subsidence in the central and eastern of NCP**
Land subsidence has been commonly observed in the NCP, and has become the main factor



that impacts regional sustainable economic and social development [Guo et al., 2015]. Over
the past years, the scope and magnitude of land subsidence has expanded. In this study, we
used GPS sites to obtain time series of land subsidence evolution characteristics. The trend
rates from GPS sites, after removing the rates from GRACE-derived long-term uplift and GIA
effects can be seen in Figure 10b (the gray background areas in the dashed white box) to
reflect the rate of land subsidence from 2010 to 2013, which is because the groundwater
exploitation in the deep confined aquifers has a more serious impact on land subsidence [Guo
et al., 2015]. The results show that Tianjin becomes the most serious subsidence area, e.g., in
the Tanggu and Hangu district (TJBH), with an average subsidence of ~14 mm/yr after 2010;
In Wuqing district (TJWQ), recent subsidence averaged ~43 mm/yr. Because the Cangxian
district (HECX) is close in proximity to the Jinghai region of Tijian, the sedimentation rate of
~20 mm/yr can represent the subsidence trend of southwest Tianjin. However, the difference
of spatial distribution of land subsidence is large in Tianjin, and uneven settlement
characteristics are obvious. For example, the southwest and western areas of Tianjin are the
most serious areas, and the trend of land subsidence exists in the northward but the amplitude
is small relative to the southwest and western areas, i.e., (JIXN site shows a small negative
trend (~-0.6 mm/yr). The cause for subsidence in the Tianjin area is linked to over
exploitation of groundwater, an issue that has not been effectively controlled resulting in
rapidly developing land subsidence in the suburbs in recent years [Yi et al., 2011].

In the central and eastern region of NCP, where disastrous land subsidence has also occurred
in Beijing and cities in central of Hebei province and the northeast of Shandong province, for





instance, large subsidence zone in Hebei province has formed from north to south, where start
from the western region of Beijing (BJFS, BJSH and BJYQ station), via the eastern region of
Xingtai and Handan (HELY station), extend to the northern of Hebi (HAHB station belong to
Henan province).

However, results from our investigation show that the center of land subsidence does not
completely overlap the TWS loss contributions (see the secular trend maps of the TWS
changes of NCP in Figure 2). The uplift still exists even when we removed the rates from
GRACE-derived and GIA effects in the piedmont of Taihang Mountains and the western part
of NCP (Shanxi province), where the groundwater depletion occurs in the shallow unconfined
aquifers have not led to a large area of subsidence. The reason for this difference with the
western region of NCP is that crustal uplift is mainly controlled by tectonic movement, which
is the orogenic belt and plateau area in western of the Taihang Mountains basic in the uplift.
In our results, most of the corrected vertical velocities at GPS stations, especially in the
central and eastern region of NCP, agree with the previous study results, i.e., combining with
mobile and continuous GPS observation [Zhao et al., 2014] and using GPS stations from
GNSS and leveling data [MLR, 2015], The results of vertical crust movement in the NCP
from the previous study, please see the Figure S6 in the supporting information.

**6. Conclusions**
Temporal variations in the geographic distribution of surface mass (continental water, ocean
mass and atmospheric mass) can lead to displacement of the Earth's surface. Due to excessive





exploitation of groundwater resources the NCP area has become susceptible to land
subsidence, and it has become one of the most affected areas in the world. Calculating the
loading displacement can explain the natural displacement phenomenon, and it presents new
insight into the dynamics of land subsidence.

Traditional displacement observation has space limitations. Based on the elastic displacement
of the Earth's crust by surface loadings, this study combined GRACE and GPS data to
investigate vertical displacements in the NCP area. GRACE data was used to model vertical
displacements due to changes in hydrological loads. The results showed both GPS and
GRACE data to observe strong seasonal variations. Comparisons between the observed GPS
seasonal vertical displacement and GRACE-derived seasonal displacement demonstrated that
a consistent physical mechanism is responsible for TWS changes, i.e., the seasonal
hydrospheric mass movements due to climate variability cause periodic displacements of the
lithosphere.

As well as the significant seasonal characteristics, GRACE also exhibited a long-term mass
loss in this region; the rate of GRACE-derived TWS loss (after multiplying by the scaling
factor) in the NCP was 3.39 cm/yr from 2003 to 2009, which is equivalent to a volume of
12.42 $km^3$/yr. The rate was 2.57 cm/yr from 2003 to 2012, equivalent to a volume of 9.41
$km^3$/yr. The TWS loss was principally due to groundwater depletion in the NCP. We
calculated that the consequent trend rate caused by the load mass change using GRACE data
and removed this hydrological effect from observed GPS vertical rates. Secular displacement



results showed that TWS losses reduced loading displacement to some extent, but the trend
rates disagree due to the difference of spatial distribution with anthropogenic depletion of
TWS in the NCP.

Particularly, land subsidence has been affecting the central and eastern region of NCP,
especially in Tianjin for the past years. Over-pumping of groundwater is the main cause of
land subsidence which has led to comprehensive detrimental effects on the society, the
economy and the natural environment. The impact of groundwater exploitation in different
aquifer systems and active faults in the different regions on land subsidence needs to be
analyzed in future investigations. For example, using GRACE to remove mass loading signals
from a GPS record requires either confidence that there is no concentrated load signal very
near the site, or a scaling factor based on a reliable model of the mass change (the
groundwater depletion rate estimated from monitoring well stations) pattern around the site.

**Acknowledgments.** The GPS data of CMONOC and IGS were made by First Crust
Monitoring    and    Application    Center,    China    Earthquake    Administration
(http://www.eqdsc.com/data/pgv-sjxl.htm). We also thank the Center of Space Research (CSR)
teams    for    their    online    accessible    GRACE    solutions
(ftp://podaac.jpl.nasa.gov/allData/grace/L2/CSR/RL05/) and the GLDAS/Noah model data
provided by the NASA Goddard Earth Sciences Data and Information Services Center
(http://disc.sci.gsfc.nasa.gov/). This work was initiated while L. Wang was visiting Prof. John
Wahr at the Department of Physics, University of Colorado at Boulder. We are grateful to Prof.



John Wahr for his helpful suggestions, including estimating the TWS changes averaged over
the NCP using averaging kernel method and loading responses due to water change using
GRACE Stokes coefficients to calculate the elastic displacement. This work is supported by
the National Natural Science Foundation of China (NSFC) (Grant No. 41504065), China
Postdoctoral Science Foundation funded project (Grant No. 2014T70753), the China
University of Geosciences (CUG) Hubei Subsurface Multi-scale Imaging Lab (Grant No.
SMIL-2014-09) and Hubei province natural science foundation of China (Grant No.
2014CFB170).

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

35−39 (in Chinese with an English abstract).





**Table captions:**
**Table 1.** GPS Station information.

**Table 2.** Trends of TWS derived by GRACE and scaled GRACE during 2003–2009 and
2003–2012.



**Figure captions:**
**Figure 1.** Study region of North Plain China (NCP) showing locations of continuous GPS
stations. White dots represent continuous GPS sites in the Crustal Movement Observation
Network of China (CMONOC) and red stars represent the International GNSS Service (IGS)
sites). Cities and provinces are labeled as follows: Beijing (BJ), Tianjin (TJ), Hebei province
(HE), and Shanxi province (SX).

**Figure 2.** The 2003–2012 secular trend maps (cm/yr) of the terrestrial water storage (TWS)
changes in North Plain China (NCP) and surrounding regions derived from GRACE data.
Results have been destriped and smoothed with a 250-km Gaussian smoothing function.

**Figure 3.** Daily values of the vertical (positive upward) components of position, as measured
at IGS GPS sites BJFS, ZHNZ, BJSH, JIXN and TAIN. The example of displacement due to
atmospheric and non-tidal ocean loading at BJFS IGS sites are shown in (b).

**Figure 4.** Surface horizontal (north and east components) and vertical deformation modeled
by GRACE in four IGS sites. (a) and (b) show the time series and trend rates of north and east
components in BJFS, JIXN, TAIN and ZHNZ, respectively, (c) show the time series of
vertical displacements.

**Figure 5.** Time series showing daily values (a) and fitting results (b) of the vertical (positive
upward) components from GPS and GRACE-derived at five IGS GPS sites.






**Figure 6.** Comparison of annual amplitudes and initial phases between GPS (grey) and
GRACE (red). The initial phases are counterclockwise from the east (reference time is

814     2004.0).


**Figure 7.** Time series showing total terrestrial water storage (TWS) changes in the spatially
averaged area (kernel) of the NCP estimated from CSR GRACE data. The dashed curve is the
temporal smoothing result.

**Figure 8.** GRACE-derived smoothed (dash curves) and long-term (solid curves) vertical
displacement time series due to load changes (a), the groundwater depletion contributions
estimated from GRACE minus GLDAS data for smoothed (dash curves) and long-term (solid
curves) vertical displacements (b), as measured at 25 GPS sites in NCP and its surrounding
region. The grey background highlight part shows inflexion effects due to TWS changes in
the vertical component.

**Figure 9.** Smoothed (dash curves) and long-term (solid curves) versions of daily values of the
vertical (positive upward) component of position, as measured at 29 GPS sites in NCP and its
surrounding region, (a) 5 IGS stations and (b) 24 CMONOC stations.

**Figure 10.** GPS (gray arrow, positive upward) and corrected GPS (red arrow, positive upward)
vertical trend rate after subtracting the GRACE-derived long-term uplift rate due to load





changes and GIA effect between 2004 and 2009 (a), and between 2010 and 2013 (b).





**Table 1:**

| Stations | Lat. | Lon. | Time | Annual Amplitude of vertical displacement (mm) | | Annual Phase of vertical displacement (days) Reference time is 2004.0 | | Trend Rates of vertical displacement (mm/yr) 2004~2009 | | Trend Rates of vertical displacement (mm/yr) 2010~2013 | |
|---|---|---|---|---|---|---|---|---|---|---|---|
| | | | | GPS | GRACE | GPS | GRACE | GPS | GRACE | GPS | GRACE |
| BJFS[*] | 39.6 | 115.8 | 2003 ~2013 | 2.50±0.26 | 1.35±0.24 | 40.09±6.39 | 359.63±10.73 | 1.47±0.14 | 0.58±0.06 | -0.37±0.23 | 0.26±0.13 |
| BJSH[*] | 40.2 | 116.2 | | 3.25±0.23 | 1.25±0.23 | 52.75±4.20 | 1.00±11.09 | 0.12±0.12 | 0.53±0.06 | -0.94±0.25 | 0.14±0.13 |
| JIXN[*] | 40 | 117.5 | | 2.46±0.22 | 1.32±0.23 | 32.20±5.11 | 359.72±10.56 | 1.21±0.12 | 0.53±0.06 | -0.19±0.20 | 0.09±0.13 |
| TAIN[*] | 36.2 | 117.1 | | 3.31±0.32 | 2.07±0.39 | 16.44±5.66 | 349.43±11.36 | 0.18±0.15 | 0.80±0.09 | 0.46±0.31 | 0.04±0.16 |
| ZHNZ[*] | 34.5 | 113.1 | | 2.38±0.36 | 2.24±0.43 | 28.18±8.94 | 354.76±11.39 | 0.04±0.15 | 0.65±0.10 | 2.55±0.32 | -0.35±0.17 |
| BJGB[#] | 40.6 | 117.1 | | 3.61±0.41 | 1.25±0.23 | 32.07±6.58 | 4.25±11.11 | | 0.49±0.06 | 0.25±0.34 | 0.02±0.13 |
| BJYQ[#] | 40.3 | 115.9 | | 3.55±0.41 | 1.23±0.23 | 25.82±6.87 | 3.34±11.26 | | 0.52±0.06 | -0.41±0.34 | 0.14±0.13 |
| HAHB[#] | 35.6 | 114.5 | | 3.44±0.42 | 2.13±0.42 | 24.12±6.93 | 349.19±11.79 | | 0.77±0.10 | -0.55±0.27 | -0.28±0.17 |
| HAJY[#] | 35.1 | 112.4 | | 2.28±0.51 | 2.05±0.44 | 9.28±13.00 | 355.65±12.80 | | 0.84±0.10 | -0.30±0.33 | -0.40±0.17 |
| HECC[#] | 40.8 | 115.8 | | 2.49±0.39 | 1.17±0.23 | 9.25±9.05 | 11.15±11.81 | | 0.49±0.06 | 1.32±0.25 | 0.06±0.13 |
| HECD[#] | 41 | 117.9 | | 4.09±0.36 | 1.27±0.23 | 40.28±5.21 | 7.63±11.09 | | 0.45±0.06 | -0.68±0.25 | -0.07±0.13 |
| HECX[#] | 38.4 | 116.9 | | 4.57±0.51 | 1.62±0.29 | 41.20±6.82 | 348.56±10.80 | | 0.73±0.07 | -20.85±0.38 | 0.26±0.14 |
| HELQ[#] | 38.2 | 114.3 | | 2.06±0.37 | 1.67±0.28 | 30.89±10.63 | 354.34±9.95 | | 0.68±0.07 | 1.76±0.26 | 0.31±0.14 |
| HELY[#] | 37.3 | 114.7 | | 2.65±0.37 | 1.88±0.33 | 10.54±8.44 | 350.61±10.56 | | 0.79±0.08 | 0.30±0.27 | 0.11±0.15 |
| HETS[#] | 39.7 | 118.2 | | 1.59±0.42 | 1.39±0.24 | 279.75±15.31 | 355.30±10.28 | | 0.53±0.06 | 3.70±0.32 | 0.07±0.13 |
| HEYY[#] | 40.1 | 114.1 | | 3.40±0.39 | 1.20±0.23 | 2.92±6.44 | 7.32±11.44 | | 0.49±0.06 | 0.76±0.30 | 0.29±0.13 |
| HEZJ[#] | 40.8 | 114.9 | 2010 ~2013 | 1.95±0.35 | 1.13±0.23 | 364.91±10.74 | 15.62±12.11 | | 0.47±0.06 | 1.06±0.25 | 0.11±0.13 |
| NMTK[#] | 40.2 | 111.2 | | 3.37±0.47 | 1.02±0.23 | 8.85±7.09 | 25.83±13.67 | | 0.37±0.06 | 1.20±0.37 | 0.51±0.13 |
| NMZL[#] | 42.2 | 115.9 | | 1.72±0.39 | 1.15±0.23 | 37.43±14.85 | 30.16±12.06 | | 0.41±0.06 | -0.49±0.28 | -0.10±0.13 |
| SDJX[#] | 35.4 | 116.3 | | 2.36±0.40 | 2.22±0.41 | 39.05±9.97 | 350.76±11.00 | | 0.71±0.10 | 0.78±0.33 | -0.07±0.17 |
| SDZB[#] | 36.8 | 117.9 | | 3.59±0.44 | 1.90±0.37 | 25.65±6.87 | 347.50±11.50 | | 0.79±0.09 | -1.12±0.30 | 0.12±0.16 |
| SXCZ[#] | 36.2 | 113.1 | | 3.99±0.47 | 1.93±0.39 | 35.27±6.48 | 352.64±12.24 | | 0.87±0.09 | 0.22±0.30 | -0.22±0.16 |
| SXGX[#] | 36.2 | 111.9 | | 1.17±0.51 | 1.81±0.39 | 311.44±26.46 | 359.06±12.86 | | 0.92±0.09 | 3.17±0.38 | -0.17±0.16 |
| SXLF[#] | 36 | 111.3 | | 3.65±0.48 | 1.79±0.40 | 18.16±7.77 | 363.02±13.33 | | 0.95±0.10 | 1.21±0.31 | -0.19±0.17 |
| SXLQ[#] | 39.3 | 114 | | 3.63±0.47 | 1.38±0.24 | 25.49±7.95 | 362.70±10.45 | | 0.54±0.06 | 1.42±0.33 | 0.39±0.13 |
| SXXX[#] | 35.1 | 111.2 | | 2.93±0.54 | 1.98±0.43 | 15.09±10.44 | 363.17±13.18 | | 0.90±0.10 | 1.30±0.41 | -0.36±0.17 |
| TJBD[#] | 39.6 | 117.3 | | 3.36±0.48 | 1.38±0.24 | 25.80±7.38 | 355.65±10.54 | | 0.58±0.06 | -1.10±0.33 | 0.16±0.13 |
| TJBH[#] | 39 | 117.6 | | 6.25±0.45 | 1.49±0.26 | 42.18±9.56 | 350.10±10.60 | | 0.64±0.06 | -16.84±0.37 | 0.20±0.13 |
| TJWQ[#] | 39.3 | 117.1 | | 5.08±0.49 | 1.43±0.25 | 0.11±8.23 | 353.28±10.58 | | 0.62±0.06 | -44.46±0.45 | 0.21±0.13 |

[*]IGS sites: the observation time between 2003 and 2013.
[#]CMONOC sites: the observation time between 2010 and 2013.









**Table 2:**

| Time Span | GRACE Trend (cm/yr of the water thickness) | GRACE Scaled (×1/0.47) Trend (cm/yr of the water thickness) | GRACE Scaled (×3.6626/0.47) Trend (km$^3$/yr of the mass) |
|---|---|---|---|
| 2003~2009 | -1.62±0.39 | -3.39±0.81 | -12.42±3.15 |
| 2003~2012 | -1.23±0.23 | -2.57±0.49 | -9.41±1.79 |






**Figure 1:**

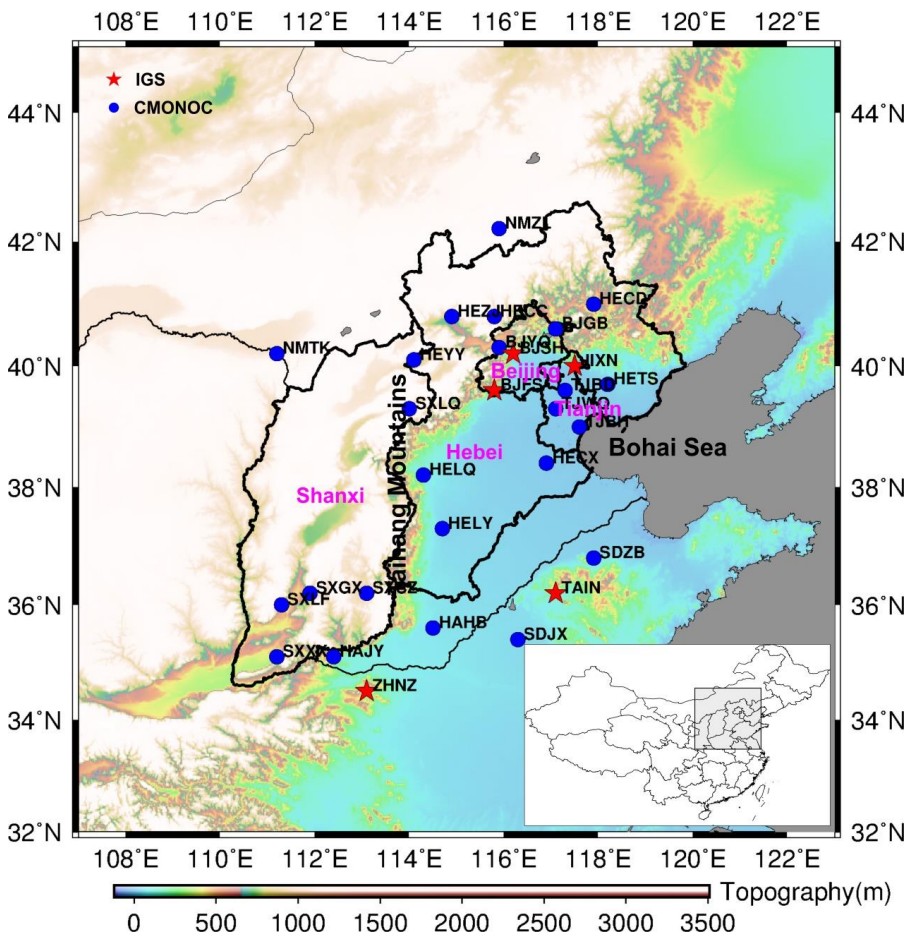





**Figure 2:**

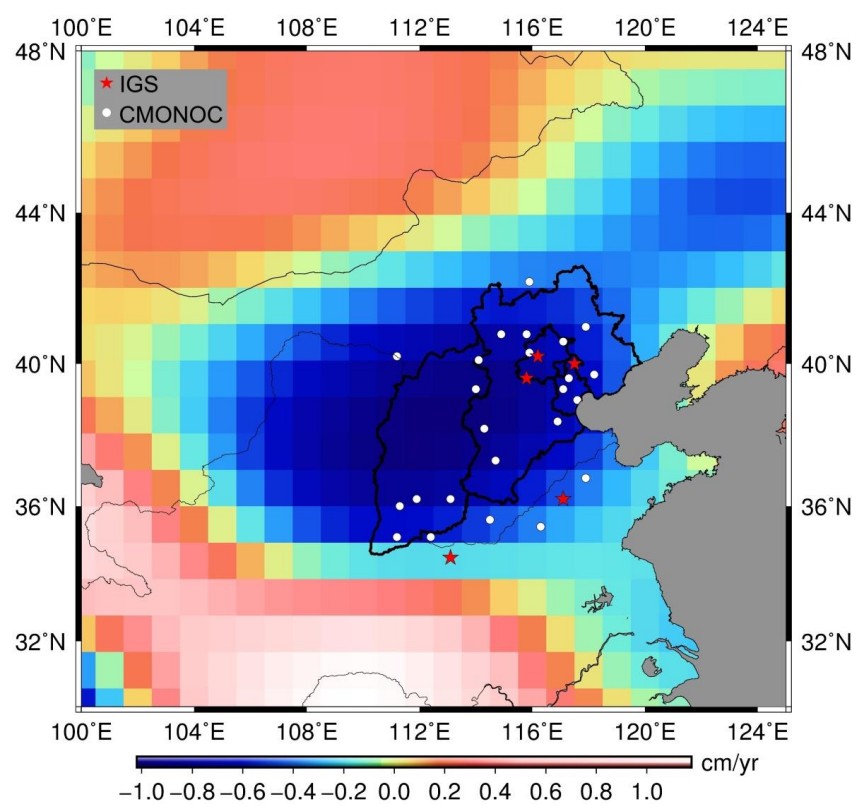






**Figure 3:**

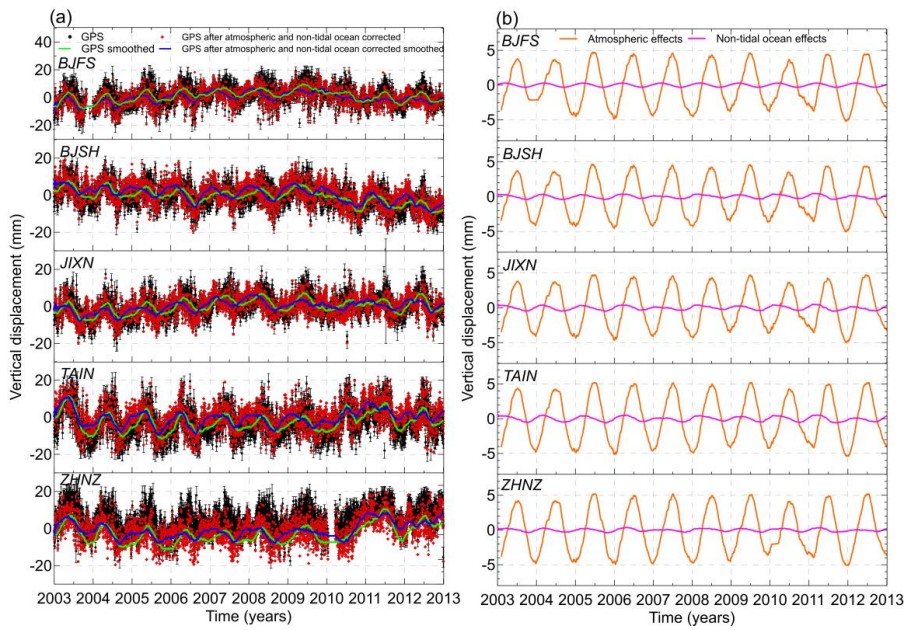




**Figure 4:**

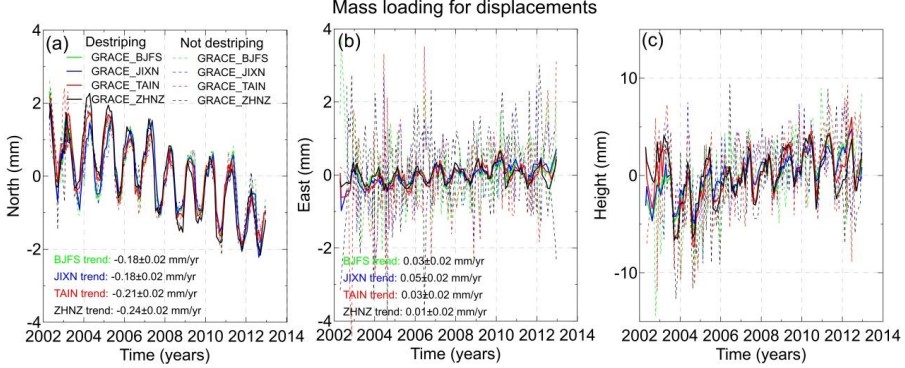




**Figure 5:**

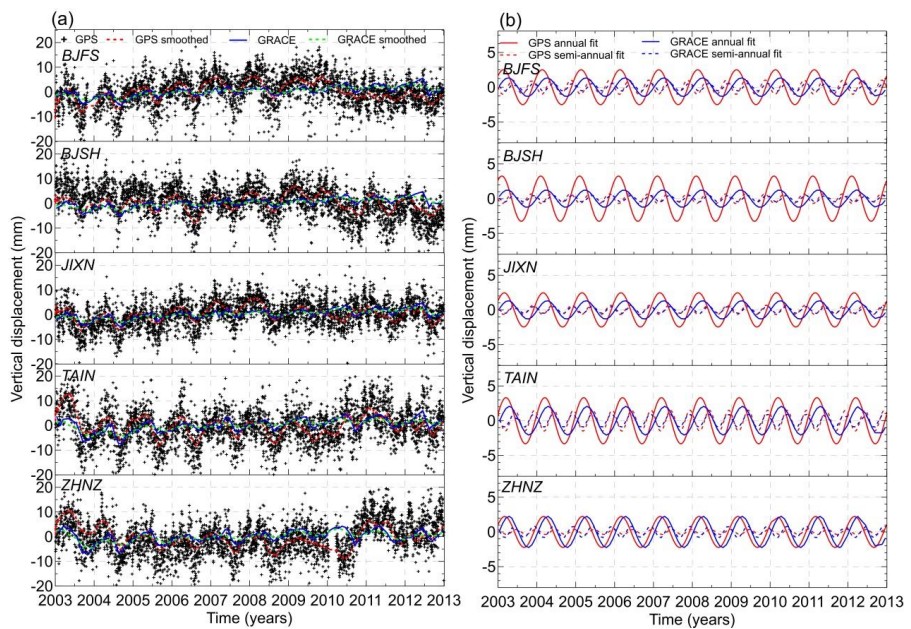






**Figure 6:**

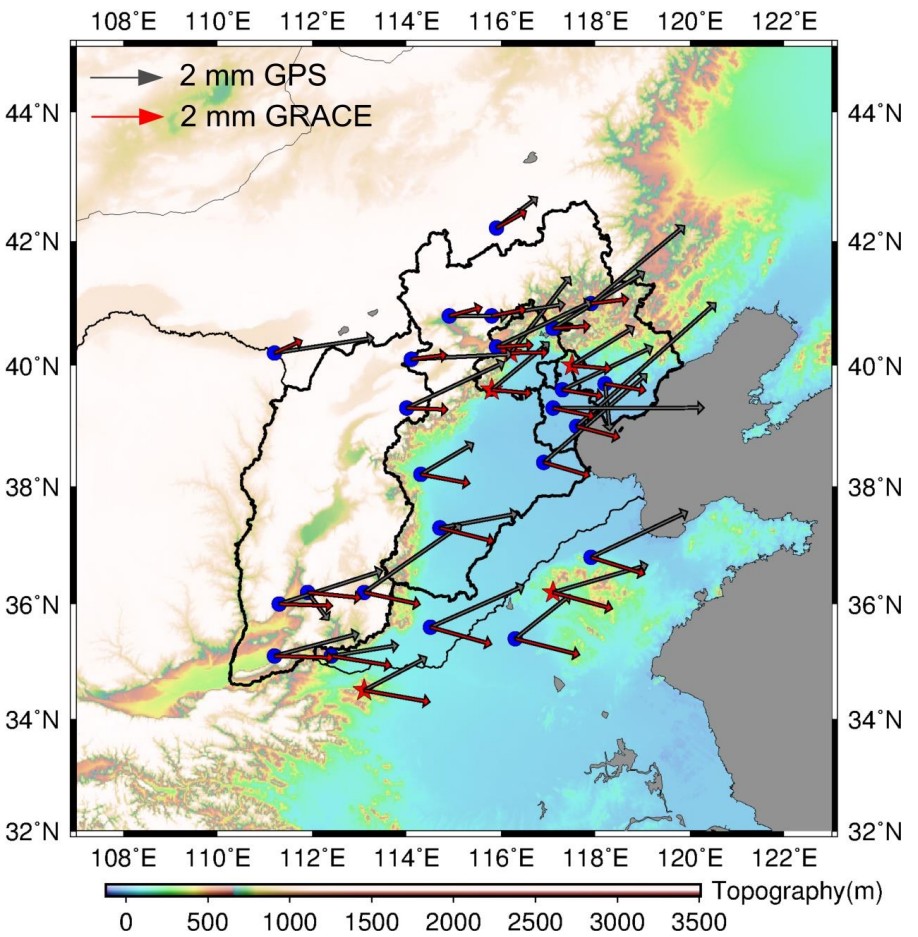





**Figure 7:**

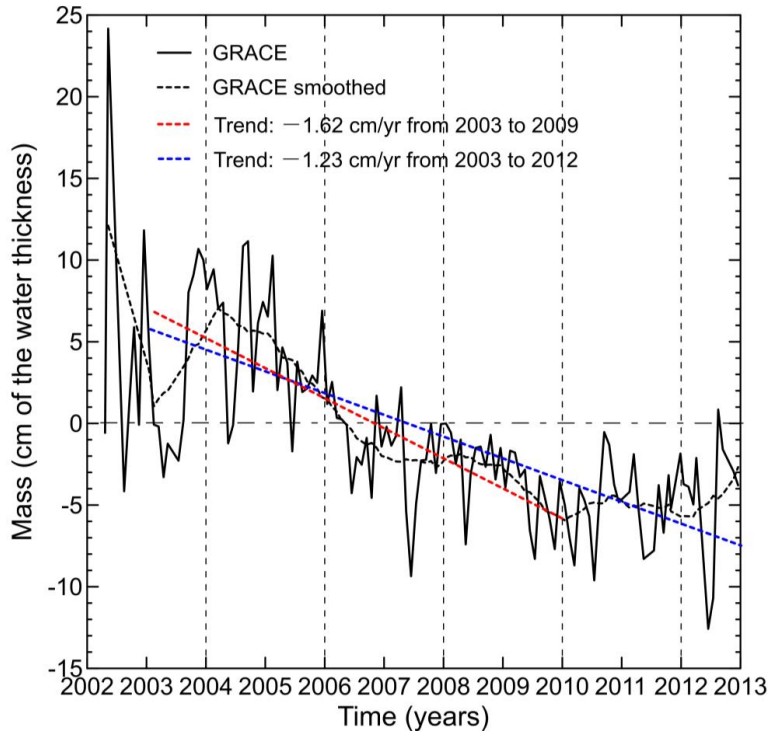







**Figure 8:**

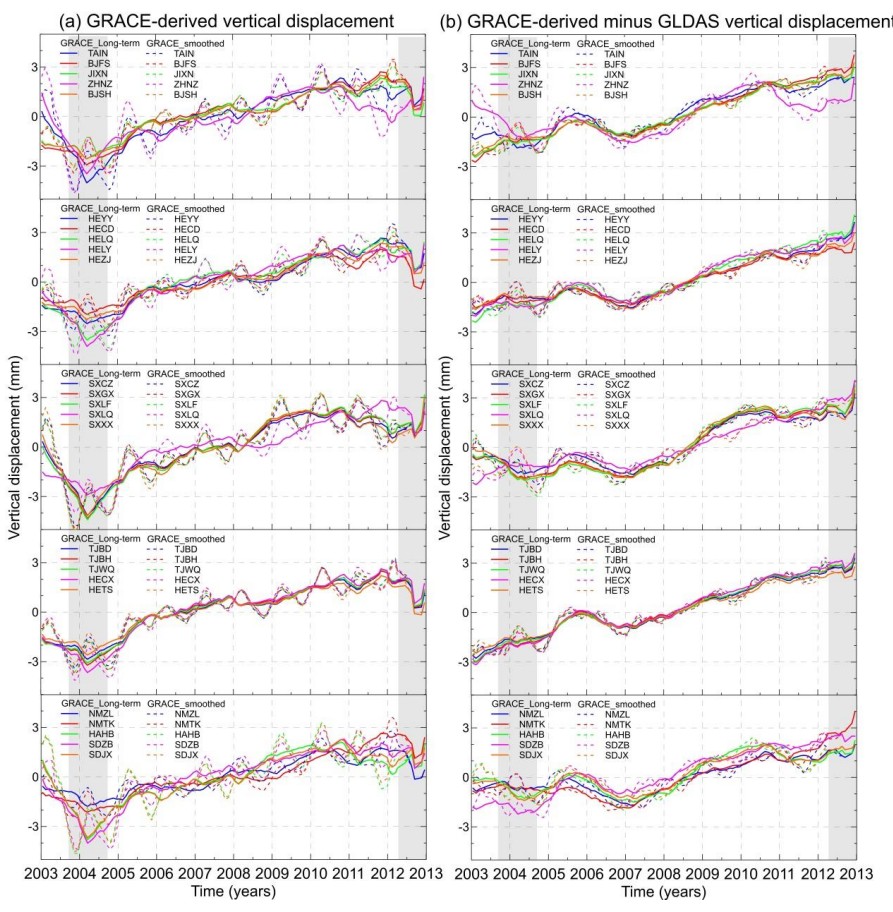






**Figure 9:**

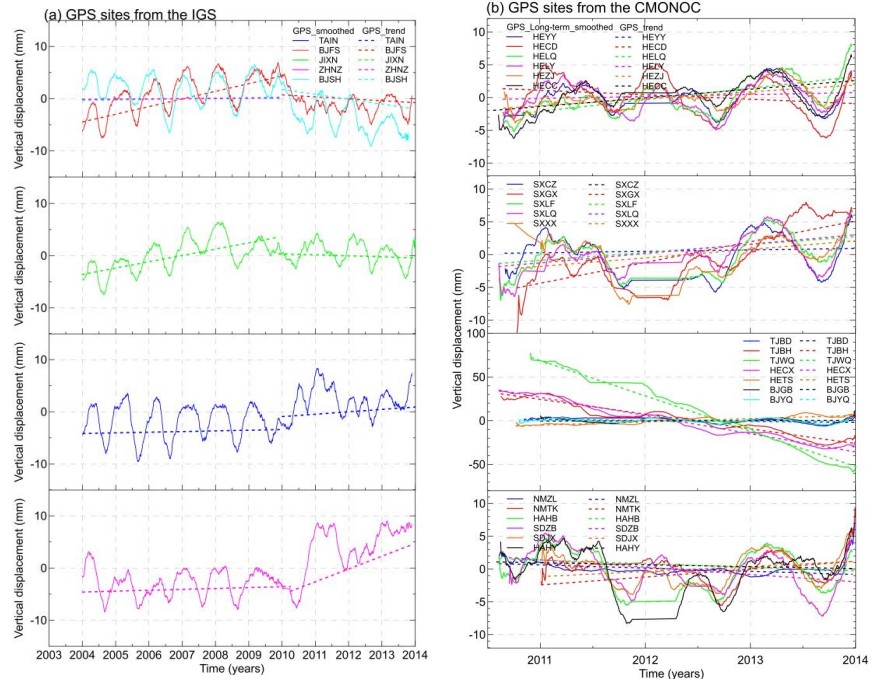




**Figure 10:**

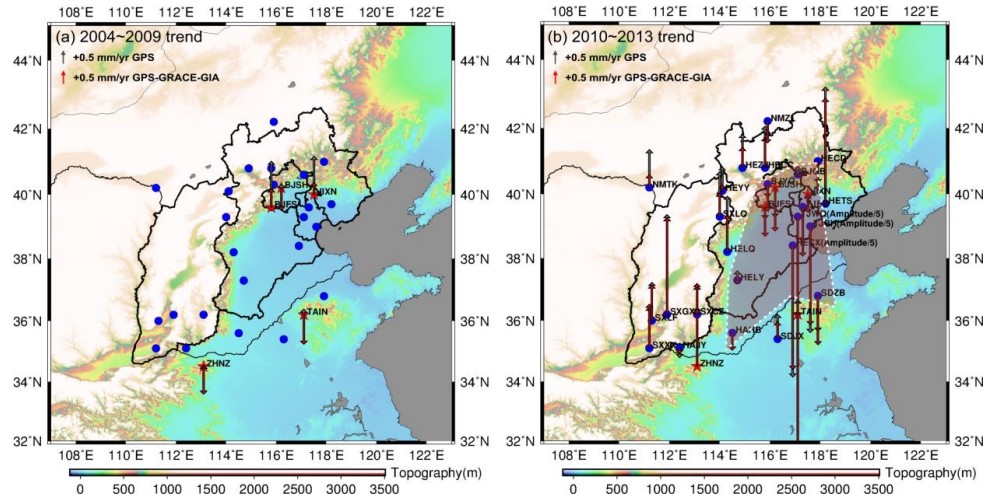
