# Peer review of "Detecting seasonal and long-term vertical displacement in the North"

_Hydrology and Earth System Sciences, 2016_

## Referee Comment (RC1) · Anonymous Referee #1 · 18 Nov 2016

In this manuscript, the authors present the time series variation of vertical displacement in North China Plain using GPS and GRACE data. Also, they analyzed the impact of Terrestrial Water Storage loss on vertical displacement. Generally, the manuscript is written clearly and illustrates some interesting results about the long-term variation of vertical displacement in North China Plain from 2003 to 2013 and discussion about the impact factors of vertical displacement like TWS loss. However, the manuscript suffers some deficiencies that need to be discussed before publishing of the manuscript. The major points that need to be added are provided below.

1) Surface vertical displacement and estimation of water storage variation using GRACE data were presented in many researches during recent years and the methodology are more or less the same. However, there is little discussion in validation of the result. In this study, is there any field measurement data of groundwater level for validation of the water storage loss in NCP?

2) In part 5.2, GPS trend changes of water storage are different in two periods (2004-2009 and 2010- 2013), especially the long-term trend rate is different in different areas from 2010 to 2013. Are there any field measurement data of groundwater levels or groundwater use in different areas can support this result? In my opinion, the manuscript would be more improved if some groundwater data can be combined into this research.

Please also note the supplement to this comment:
http://www.hydrol-earth-syst-sci-discuss.net/hess-2016-552/hess-2016-552-RC1-supplement.pdf

---

## Referee Comment (RC2) · Anonymous Referee #2 · 13 Dec 2016

General Comments:

This manuscript investigated the seasonal displacements of surface loadings in the NCP area using GRACE and GPS. The consistency between GPS and GRACE was quantitatively evaluated by removing GRACE-derived seasonal displacement from GPS observed detrended height time series. The rate of GRACE-derived TWS loss in the NCP was estimated and the land subsidence in the central and eastern of NCP was discussed. Generally the topic of manuscript is interesting and the writing is clear. However, in my opinion there are several important issues need to be addressed before publication.

Major points:

1. Recently lots of publications regarded on the estimation of TWS variation in the NCP, such as Huang et al. (2015), Feng et al. (2013), Moiwo et al. (2013), Tang et al. (2013) (listed in the manuscript) and some other references (e.g. Su et al., 2011). These studies should be included in the discussion to compare the depletion in TWS in the NCP.

2. Results and discussion should be presented separately, since some result sections (e.g. section 3) include discussion parts (e.g. lines 308-326, lines 356-359). I would suggest moving the relevant parts to section 5 for discussion.

3. There are some grammatical errors in the paper, meanwhile some sentences are not well written (e.g., lines 301-302, lines 335-338, lines 422-424, line 448). Please check it carefully throughout the manuscript.

Specific Comments:

1. Page 1, line 15: please remove "We employ" from the sentence.

2. Page 1, lines 17 and 19: please give the full name of "NCP" and "WRMS", since it first appeared in the paper.

3. Line 367: should be "equations (2) and (3)".

4. Line 369: » "was presented..."

5. Line 388: » "under- or overestimation..."

6. Line 449: » "before and after removing..."

7. Line 488: "two periods" » "two sub-periods"

References:

Su, X. L., J. S. Ping, and Q. X. Ye (2011), Terrestrial water variations in the North China Plain revealed by the GRACE mission, Sci. China Earth Sci., 54 (12), 1965–1970, doi:10.1007/s11430-011-4280-4.

---

## Author Comment (AC1) · 25 Dec 2016

Dear Anonymous Referee #1,

Thanks very much for your constructive comments concerning our manuscript entitled "Detecting seasonal and long-term vertical deformation in the North China Plain using GRACE and GPS". Those comments are all valuable and very helpful into revising and improving our paper, as well as the important guiding significance to our researches. We have studied comments carefully and here replied each comment bellow. The original comments are in plain text and the replies in italics which we hope meet with approval.

**Anonymous Referee #1 ()**

In this manuscript, the authors present the time series variation of vertical displacement in North China Plain using GPS and GRACE data. Also, they analyzed the impact of Terrestrial Water Storage loss on vertical displacement. Generally, the manuscript is written clearly and illustrates some interesting results about the long-term variation of vertical displacement in North China Plain from 2003 to 2013 and discussion about the impact factors of vertical displacement like TWS loss. However, the manuscript suffers some deficiencies that need to be discussed before publishing the manuscript. The major points that need to be added are provided below.

1) Surface vertical displacement and estimation of water storage variation using GRACE data were presented in many researches during recent years and the methodology are more or less the same. However, there is little discussion in validation of the result. In this study, is there any field measurement data of groundwater level for validation of the water storage loss in NCP?

*>Linsong Wang et al.: Thank you very much for the good comments. You mentioned that our result needs more discussions based on measurement data of groundwater level in NCP. We have acquired in situ groundwater level measurements (most of groundwater table depth in the shallow unconfined aquifers, available from 2002 to 2013), which mainly locating in the central and eastern plain of the NCP (including the Beijing, Tianjin and some cities of Hebei, Henan and Shandong province). The data series are obtained from Ministry of Water Resources of China (available at: http://sqqx.hydroinfo.gov.cn/shuiziyuan/). In the revised manuscript, we will get the area-weighted mean groundwater level change series in the NCP from time series of monthly groundwater table depth changes of 20 cities in our study region. Using the mean time series of*

*monthly groundwater level data, we can do a comparison between monthly groundwater storage (GWS) variations estimated from GRACE minus GLDAS/Noah model and monthly groundwater level changes observed by monitoring wells after multiplying by mean value of specific yields in the NCP during 2002-2012 (Fig. R1). Besides, according to the good comments of the Referee #2, we are also going to compare the depletion in TWS or GWS between our results and previous studied results (e.g., Huang et al., 2015; Feng et al., 2013; Moiwo et al., 2013; Tang et al., 2013; Su et al., 2011) in the NCP. We believe these discussions can validate the GWS loss in the NCP.*

[Figure]

*Figure R1. Time series showing total terrestrial water storage (TWS) changes in the spatially averaged area (kernel) of the NCP estimated from GRACE data and groundwater storage (GWS) variations from monthly groundwater level changes observed by monitoring wells.*

2) In part 5.2, GPS trend changes of water storage are different in two periods (2004-2009 and 2010-2013), especially the long-term trend rate is different in different areas from 2010 to 2013. Are there any field measurement data of groundwater levels or groundwater used in different areas can support this result? In my opinion, the manuscript would be more improved if some groundwater data can be combined into this research.

*> Linsong Wang et al.: Thank you again. Before response this comment, we would like to explain why using GPS data after removed GRACE-derived deformations to detect long-term vertical deformation in the NCP. The detailed reasons are as follows.*

*1) The GPS is now widely used in the geosciences for the estimation of precise station coordinates. The placement of a load on the Earth's surface causes deformation of the underlying solid Earth and displacements of its surface. GPS measurements of those displacements can provide information about the load.*

*2) Some previous studies have focused on the vertical component of crustal motion. These researches rely on the accurate interpretation of GPS motion in terms of surface stress or tectonic movement, and the deformation signal from surface mass loading is a source of noise. In these applications, they would like to obtain reliable loading models or even surface mass observations, which can be used to reduce the environmental loading contributions to the GPS observations. For some surface loads, such as the atmosphere, the loads are currently modeled to a fairly high degree of accuracy (van Dam et al., 1994). However, for other loads, especially the distribution of water mass on continents (soil moisture, groundwater, snow and ice) the load is poorly known in most regions of the globe, but the deformation it causes is large enough to contribute to the GPS signal (van Dam et al., 2001). Fortunately, with the development of GRACE, the mass variations from hydrologic loading now can be quantitatively estimated. The GRACE-derived time-variable gravity field coefficients can be converted to harmonic coefficients for crustal deformation in three components.*

*Next, please see our detailed responses below.*

*1) As shown in Figure 8a in our manuscript, the obvious uplift are presented in the decomposition of the signal based on the GRACE-derived TWS trend, which showed the long-term mass loss in the NCP from 2003 to 2009, but the rate of this decrease slowed towards the end of 2009 and then increases again after 2010. This may explain the differences of GPS trend changes of water storage in these two sub-periods. In addition, in situ groundwater level measurements also confirmed the difference of trend changes of GWS in these two sub-periods that you mentioned "GPS trend changes of water storage are different in two sub-periods (2004-2009 and 2010-2013)" (Fig. R1).*

*2) However, the water mass loss trend from GRACE or in situ data is inconsistent with the GPS results in our study, which you mentioned 'especially the long-term trend rate is different in different areas from 2010 to 2013'. The groundwater level changes from observations of shallow aquifers shows a persistent increase after 2010, when the annual precipitation begins to increase (Fig. R1). Obviously, the recharges from precipitation for deep aquifers are different with the shallow aquifers. In addition, these discrepancies may reflect uncertainties in GRACE, land surface models (LSMs) and groundwater monitoring well data. It also can explain the different GRACE-derived uplift caused by total GWS change (GRACE minus GLDAS/Noah) in different stations from 2010 to 2013 (Fig.8b in manuscript).*

*In fact, it is more important about the applicability of loading theory in the NCP. The previous studied results showed that the groundwater depletion occurs in the shallow unconfined aquifers in Piedmont Plain while groundwater depletion occurs in the deep confined aquifers in the central and eastern plain of the NCP. Especially in the central and eastern plain of the NCP, although some GPS can detect land subsidence due to the occurrence of groundwater depletion in the deep confined aquifers, but the loading deformation effect from mass change is still remain in GPS long-term trend. Thus, we compute the GRACE-derived long-term trend from the CSR solutions for all continuous GPS sites used in this paper. The results indicate GRACE-derived an overall uplift for the whole region at the 0.37~0.95 mm/yr level from 2004 to 2009, but the rate of change direction is inconsistent in different GPS stations at -0.40~0.51 mm/yr level from 2010 to 2013 (Table 1 in manuscript). Then we remove this hydrological-induced long-term trend from GPS actual observed vertical rates to derive the corrected vertical velocities (Figure 9 in manuscript), which can be used to study the vertical crust movement caused by single tectonic movement (fault activity or land subsidence).*

Merry Christmas and Happy New Year!

Best regards,

Linsong Wang et al.

25 December 2016

---

## Author Comment (AC2) · 25 Dec 2016

Dear Anonymous Referee #2,

Thank you for your constructive comments concerning our manuscript entitled "Detecting seasonal and long-term vertical deformation in the North China Plain using GRACE and GPS". Those comments are all valuable and very helpful for revising and improving our paper, as well as the important guiding significance to our researches. We will revise the manuscript according to your comments and here replied each comment bellow. The original comments are in plain text and the replies in italics.

**Anonymous Referee #2 ()**

**General Comments:**

This manuscript investigated the seasonal displacements of surface loadings in the NCP area using GRACE and GPS. The consistency between GPS and GRACE was quantitatively evaluated by removing GRACE-derived seasonal displacement from GPS observed detrended height time series. The rate of GRACE-derived TWS loss in the NCP was estimated and the land subsidence in the central and eastern of NCP was discussed. Generally the topic of manuscript is interesting and the writing is clear. However, in my opinion there are several important issues need to be addressed before publication.

**Major points:**

1. Recently lots of publications regarded on the estimation of TWS variation in the NCP, such as Huang et al. (2015), Feng et al. (2013), Moiwo et al. (2013), Tang et al. (2013) (listed in the manuscript) and some other references (e.g. Su et al., 2011). These studies should be included in the discussion to compare the depletion in TWS in the NCP.

*>Linsong Wang et al.: Thank you very much for the good comments that you mentioned previous studies should be included in the discussion. We are going to compare the depletion in TWS or GWS between our results and previous studied results (i.e., add a table including mass loss trend from our results and previous studied in the revised manuscript) in the NCP (e.g., the reported TWS loss from Zhong et al.(2009), Su et al.(2011), Moiwo et al.(2009), and the reported GWS loss from Huang et al.(2015), Feng et al.(2013), Tang et al.(2013)). Besides, according to the good comments of the Referee #1, we are also going to compare the depletion in GWS between monthly*

*GWS variations estimated from GRACE minus GLDAS/Noah model and monthly groundwater level changes observed by monitoring wells during 2002-2012.*

2. Results and discussion should be presented separately, since some result sections (e.g. section 3) include discussion parts (e.g. lines 308-326, lines 356-359). I would suggest moving the relevant parts to section 5 for discussion.

*>Linsong Wang et al.: Thanks for your good suggestion. We will move the discussion parts in section 3 (e.g. lines 308-326, lines 356-359) to section 5 for discussing the cause of the difference between our results and previous studies.*

3. There are some grammatical errors in the paper, meanwhile some sentences are not well written (e.g., lines 301-302, lines 335-338, lines 422-424, line 448). Please check it carefully throughout the manuscript.

*>Linsong Wang et al.: Thank you again. We will rewrite your mentioned some sentences and continue to polish the manuscript.*

**Specific Comments:**

1. Page 1, line 15: please remove "We employ" from the sentence.

*>Linsong Wang et al.: Thank you. We will remove "We employ" from the sentence in the revised manuscript.*

2. Page 1, lines 17 and 19: please give the full name of "NCP" and "WRMS", since it first appeared in the paper.

*>Linsong Wang et al.: Thank you. We will give the full name of "NCP" and "WRMS" where they first appear in the paper.*

3. Line 367: should be "equations (2) and (3)".

*>Linsong Wang et al.: Thank you for reminding our negligence.*

4. Line 369: » "was presented: : :"

*>Linsong Wang et al.: Thank you. "were presented" will be changed to "was presented" in the revised manuscript.*

5. Line 388: » "under- or overestimation: : :"

*>Linsong Wang et al.: Thank you. "under or overestimation" will be revised to "under- or overestimation" in the revised manuscript.*

6. Line 449: » "before and after removing: : :"

>*Linsong Wang et al.: Thank you. "before and after remove" will be changed to "before and after removing" in the revised manuscript.*

7. Line 488: "two periods" » "two sub-periods"

>*Linsong Wang et al.: Thank you. "two periods" will be changed to "two sub-periods" in the revised manuscript.*

References:

Su, X. L., J. S. Ping, and Q. X. Ye (2011), Terrestrial water variations in the North China Plain revealed by the GRACE mission, Sci. China Earth Sci., 54 (12), 1965–

1970, doi:10.1007/s11430-011-4280-4.

>*Linsong Wang et al.: Thank you for providing the reference of previous studies.*

Merry Christmas and Happy New Year!
Best regards,

Linsong Wang et al.

25 December 2016

---

## Author Response (AR1)

**Dear Editors and Reviewers:**

Thanks very much for your letter and for the reviewers' comments concerning our manuscript entitled "Detecting seasonal and long-term vertical deformation in the North China Plain using GRACE and GPS" (No. hess-2016-552). Those comments are all valuable and very helpful for revising and improving our paper, as well as the important guiding significance to our researches. We have studied comments carefully and have made major revision which we hope to meet with approval. Revised portion are marked in the "Marked-up manuscript version". The main corrections in the paper and the responds to the reviewer's comments are as following.

**PartⅠ. Responses to the comments**

**Anonymous Referee #1 ()**

In this manuscript, the authors present the time series variation of vertical displacement in North China Plain using GPS and GRACE data. Also, they analyzed the impact of Terrestrial Water Storage loss on vertical displacement. Generally, the manuscript is written clearly and illustrates some interesting results about the long-term variation of vertical displacement in North China Plain from 2003 to 2013 and discussion about the impact factors of vertical displacement like TWS loss. However, the manuscript suffers some deficiencies that need to be discussed before publishing of the manuscript. The major points that need to be added are provided below.

**Dear Anonymous Referee #1,**

**Thanks very much for your constructive comments. We have studied comments carefully and here replied each comment bellow. The original comments are in plain text and the replies in italics which we hope meet with approval.**

1) Surface vertical displacement and estimation of water storage variation using GRACE data were presented in many researches during recent years and the methodology are more or less the same. However, there is little discussion in validation of the result. In this study, is there any field measurement data of groundwater level for validation of the water storage loss in NCP?

*>Linsong Wang et al.: Thank you very much for the good comments that you mentioned our study result need more discuss, basing on measurement data of groundwater level in NCP. We have acquired in-situ groundwater level measurements (available only from 2002 to 2013), which are mainly located in the central and eastern plain of NCP (including the Beijing and Tianjin city, some cities of Hebei Henan and Shandong province) and are obtained from Ministry of Water Resources of China (available at: http://sqqx.hydroinfo.gov.cn/shuiziyuan/). We also collected the daily precipitation data (rainfall amount) for weather stations during the period of 2003–2012 from China Meteorological Data Sharing Service System (CMDSSS) (available at: http://cdc.cma.gov.cn/index.jsp). In revised manuscript, we have get the*

*area-weighted mean groundwater level change series in the NCP from time series of monthly groundwater table depth changes of 20 cities in our study region (Figure 7 in revised manuscript). Using the mean time series of monthly groundwater level data, we can compare the rate of loss between groundwater storage (GWS) variations estimated from the data that GRACE minus GLDAS/Noah model and monthly groundwater level changes observed by monitoring wells during 2002-2012 (Table 2 in revised manuscript). Besides, according to the good comments of the Referee #2, we have compared the depletion in TWS or GWS between our results and previous studied results (e.g., Huang et al., 2015; Feng et al., 2013; Moiwo et al., 2013; Tang et al., 2013; Su et al., 2011) in the NCP (Table 2 in revised manuscript). The details of compare groundwater storage (GWS) variations with in-situ measurements and previous results, please see Section 4.1 in revised manuscript, we believe these discussions can validate GWS loss in NCP.*

2) In part 5.2, GPS trend changes of water storage are different in two periods (2004-2009 and 2010- 2013), especially the long-term trend rate is different in different areas from 2010 to 2013. Are there any field measurement data of groundwater levels or groundwater use in different areas can support this result? In my opinion, the manuscript would be more improved if some groundwater data can be combined into this research.

*> Linsong Wang et al.: Thank you again. As shown in Figure 8b in our study, the obvious uplifts are presented in the decomposition of the signal based on the*

*GRACE-derived GWS trend (GRACE minus GLDAS/Noah), which shows the long-term groundwater depletion in the NCP from 2002 to 2012. Recently, previous studied have reported the TWS loss (e.g., Zhong et al., 2009; Su et al., 2011; Moiwo et al., 2009) or GWS loss (e.g., Huang et al., 2015; Feng et al., 2013; Tang et al., 2013;) based on GRACE and land surface models (LSMs) in the NCP, these results are also consistent with that from in situ measurement data of groundwater levels. However, the water mass loss trend from GRACE and situ data show an inconsistence with the GPS results in our study, which you mentioned GPS trend changes of water storage are different in two periods (2004-2009 and 2010- 2013), especially the long-term trend rate is different in different areas from 2010 to 2013. For the reasons why the long-term trend rate from GPS is different in different areas, please see our detailed responses below.*

*1) The GPS is nowadays widely used in the geosciences for the estimation of precise station coordinates. The placement of a load on the Earth's surface causes deformation of the underlying solid Earth and displacements of its surface. GPS measurements of those displacements can provide information about the load.*

*2) Some previous studies have focused on the vertical component of crustal motion. These researches rely on the accurate interpretation of GPS motion in terms of surface stress or tectonic movement, and the deformation signal from surface mass loading is a source of noise. For these applications, they would like to obtain reliable*

*loading models or even surface mass observations, which can be used to reduce the contributions made to the GPS observations by the environmental loading. For some surface loads, such as the atmosphere, the loads are currently modeled to a fairly high degree of accuracy (van Dam et al., 1994). However, for other loads, especially the distribution of water mass on continents (soil moisture, groundwater, snow and ice), which are poorly known in most regions of the globe, but the deformation it causes is large enough to contribute to the GPS signal (van Dam et al., 2001). Fortunately, with the development of GRACE, the mass variations from hydrologic loading now can be quantitatively estimated. The GRACE-derived time-variable gravity field coefficients can be converted to harmonic coefficients for crustal deformation in three components.*

*3) In fact, it is more important to apply loading theory in the NCP. The previous studied results shown the groundwater depletion occurs in the shallow unconfined aquifers in Piedmont Plain while groundwater depletion occurs in the deep confined aquifers in the central and eastern plain of NCP. In this study, we found that GRACE-derived GWS changes are in disagreement with the groundwater level changes from observations of shallow aquifers not only from 2003 to 2009, especially from 2010 to 2013. Although the shallow groundwater can be recharged from the annual climate-driven rainfall (e.g., the groundwater level changes from observations of shallow aquifers shows a persistent increase after 2010, when the annual precipitation begins to increase), but the important facts indicate that GWS depletion*

*is more serious in deep aquifers. Especially in the central and eastern plain of NCP, although some GPS can detect land subsidence due to the occurrence of groundwater depletion in the deep confined aquifers, but the loading uplift effect from mass loss is still remain in GPS long-term trend. Thus, we compute the GRACE-derived long-term uplift using the trend from the CSR solutions for all continuous GPS sites used in this paper. The results indicate GRACE-derived data have an overall uplift in the whole region at the 0.37~0.95 mm/yr level from 2004 to 2009, but the rate of change direction is inconsistent in different GPS stations at -0.40~0.51 mm/yr level from 2010 to 2013 (Table 1). Our study indicates that the elastic responses are induced by all depth of TWS loading. Then we remove this hydrological-induced long-term trend from GPS actual observed vertical rates to derive the corrected vertical velocities (Figure 10), which can be used to study the vertical crust movement caused by tectonic movement and human activities. The results show that there are uplift areas and subsidence areas in NCP. Almost the whole central and eastern region of NCP suffers serious ground subsidence, caused by the anthropogenic-induced groundwater exploitation in the deep confined aquifers. In addition, that the ground uplifts lightly in the western region of NCP is mainly controlled by tectonic movement (e.g., Moho uplifting or mantle upwelling). Here Stokes coefficients resulted from A et al. [2013] were used to remove contributions from GIA.*

**Anonymous Referee #2 ()**

**General Comments:**

This manuscript investigated the seasonal displacements of surface loadings in the NCP area using GRACE and GPS. The consistency between GPS and GRACE was quantitatively evaluated by removing GRACE-derived seasonal displacement from GPS observed detrended height time series. The rate of GRACE-derived TWS loss in the NCP was estimated and the land subsidence in the central and eastern of NCP was discussed. Generally the topic of manuscript is interesting and the writing is clear. However, in my opinion there are several important issues need to be addressed before publication.

**Dear Anonymous Referee #2,**

**Thanks very much for your constructive comments. We have studied comments carefully and here replied each comment bellow. The original comments are in plain text and the replies in italics which we hope meet with approval.**

**Major points:**

1. Recently lots of publications regarded on the estimation of TWS variation in the NCP, such as Huang et al. (2015), Feng et al. (2013), Moiwo et al. (2013), Tang et al. (2013) (listed in the manuscript) and some other references (e.g. Su et al., 2011). These studies should be included in the discussion to compare the depletion in TWS in the NCP.

*>Linsong Wang et al.: Thank you very much for the good comments that you mentioned previous studies should be included in the discussion. In revised manuscript, we have compared the depletion in TWS or GWS between our results and previous studied results (i.e., add comparisons to the Table 2 including mass loss trend from our results and previous studied in the revised manuscript) in the NCP, e.g., the reported TWS loss from Zhong et al.(2009), Su et al.(2011), Moiwo et al.(2009), and the reported GWS loss from Huang et al.(2015), Feng et al.(2013), Tang et al.(2013). Besides, according to the good comments of the Referee #1, we have also compared the groundwater depletion in NCP between GWS variations estimated from the data that GRACE minus GLDAS/Noah model and monthly groundwater level changes observed by monitoring wells during 2002-2012 (Table 2 in revised manuscript), we believe these discussions can validate GWS loss in NCP.*

2. Results and discussion should be presented separately, since some result sections (e.g. section 3) include discussion parts (e.g. lines 308-326, lines 356-359). I would suggest moving the relevant parts to section 5 for discussion.

*>Linsong Wang et al.: Thanks for your good suggestion. In revised manuscript, we have moved the discussion parts of section 5.1 "Groundwater Depletion Contributions to Long-Term Uplift" in original manuscript to the section 4.2, which is mainly to explain long-term uplift caused by TWS loss, especially the contributions from groundwater depletion in the NCP. Then, we have added new discussion in Section 5.1 "The Loading Effects of Non-tidal Ocean and Atmospheric Variations"*

*for discussing the cause of the difference between our results and previous studies, which you mentioned "some result sections (e.g. section 3) include discussion parts (e.g. lines 308-326, lines 356-359)".*

3. There are some grammatical errors in the paper, meanwhile some sentences are not well written (e.g., lines 301-302, lines 335-338, lines 422-424, line 448). Please check it carefully throughout the manuscript.

*>Linsong Wang et al.: Thank you again. We rewrote your mentioned some sentences (please see lines 317-319, lines 332-337, lines 491-497 in revised manuscript) and the revised version of the manuscript was polished by one language editor (please see the Marked manuscript).*

**Specific Comments:**

1. Page 1, line 15: please remove "We employ" from the sentence.

*>Linsong Wang et al.: Thank you. We have removed "We employ" from the sentence in the revised manuscript.*

2. Page 1, lines 17 and 19: please give the full name of "NCP" and "WRMS", since it first appeared in the paper.

*>Linsong Wang et al.: Thank you. We have given the full name of "NCP" and "WRMS" where they first appear in the paper.*

3. Line 367: should be "equations (2) and (3)".

>*Linsong Wang et al.: Thank you for reminding our negligence. The mistake has been modified.*

4. Line 369: » "was presented: : :"

>*Linsong Wang et al.: Thank you. "were presented" has been changed to "was presented" in the revised manuscript.*

5. Line 388: » "under- or overestimation: : :"

>*Linsong Wang et al.: Thank you. "under or overestimation" has been revised to "under- or overestimation" in the revised manuscript.*

6. Line 449: » "before and after removing: : :"

>*Linsong Wang et al.: Thank you. "before and after remove" has been changed to "before and after removing" in the revised manuscript.*

7. Line 488: "two periods" » "two sub-periods"

>*Linsong Wang et al.: Thank you. "two periods" has been changed to "two sub-periods" in the revised manuscript.*

**The marked manuscript:**

[revised manuscript text omitted]

**Figure 8:**

[Figure]

**Figure 9:**

[Figure]

**Figure 10:**

[Figure]

---

## Referee Report (RR1)

The authors added the analysis of time series change of the groundwater level measurements in revised manuscript and validated the water storage loss in NCP from some extent. The authors also compared the difference between the result in the manuscript and previous study. Furthermore, the reason why water mass loss trend from GRACE is different with GPS results has been explained carefully in the revised manuscript. In my opinion, the manuscript can be accepted and be published in this journal now.

---

## Referee Report (RR2)

I am pleased to see the revised manuscript has addressed most questions that I have raised for the previous version.

One minor comment: the abstract is a bit long (likely more than 450 words) that it is difficult for the readers to follow the innovation of the paper. I suggest simplifying the abstract to highlight the new findings.

Page 2, line 35: please remove the word ''but'' from the sentence.